# LARGE LANGUAGE MODELS AS NONDETERMINISTIC CAUSAL MODELS

## ABSTRACT

Recent work by Chatzi et al. and Ravfogel et al. has developed, for the first time, a method for generating counterfactuals of probabilistic Large Language Models. Such counterfactuals tell us what would – or might – have been the output of an LLM if some factual prompt $\mathbf{x}$ had been $\mathbf{x}^*$ instead. The ability to generate such counterfactuals is an important necessary step towards explaining, evaluating, and comparing, the behavior of LLMs. We argue, however, that the existing method rests on an ambiguous interpretation of LLMs: it does not interpret LLMs literally, for the method involves the assumption that one can change the implementation of an LLM's sampling process without changing the LLM itself, nor does it interpret LLMs as intended, for the method involves explicitly representing a *nondeterministic* LLM as a *deterministic* causal model. We here present a much simpler method for generating counterfactuals that is based on an LLM's intended interpretation by representing it as a nondeterministic causal model instead. The advantage of our simpler method is that it is directly applicable to any black-box LLM without modification, as it is agnostic to any implementation details. The advantage of the existing method, on the other hand, is that it directly implements the generation of a specific type of counterfactuals that is useful for certain purposes, but not for others. We clarify how both methods relate by offering a theoretical foundation for reasoning about counterfactuals in LLMs based on their intended semantics, thereby laying the groundwork for novel application-specific methods for generating counterfactuals.

## 1 INTRODUCTION

Imagine we ask a Large Language Model (LLM): "What is your favorite color?", and it replies with "Blue". We could have asked it many other closely related questions instead, for example we might have asked it: "What is your least favorite color?". Unfortunately it is impossible (for now at least) to turn back time and make it such that we did in fact ask it the second question: no matter what, it will forever and always remain the case that we find ourselves in a world where, on this specific occasion, we asked it the first question.[1] However, one might reason, this is entirely of no consequence, for we can simply ask it the second question on another occasion, and we will have our answer then. Under this view, the generation of counterfactuals – which is what we are doing when answering the second question – is entirely straightforward, as it is no different from generating standard outputs of an LLM. We refer to this view as the *simple semantics* of counterfactuals.

Everyone agrees that the simple semantics is correct if the LLM is set to behave deterministically, which is accomplished – at least approximately – by setting the "temperature" to 0. The reason is that in the deterministic setting an LLM just corresponds to some function $\mathbf{Y} = f(\mathbf{X})$, where $\mathbf{X}$ is the textual input given by the user – called the *prompt* – and $\mathbf{Y}$ is the textual output generated by the LLM. But once the LLM is set to behave nondeterministically, which is accomplished by setting a positive temperature, the simple semantics is assumed to be unsound by Chatzi et al. (2025) and Ravfogel et al. (2025). Instead, they assume that the generation of counterfactuals demands a much more complicated semantics, one that cannot be evaluated without modifying the source code of an LLM. As a consequence, we cannot generate counterfactual outputs for probabilistic LLMs for all the most popular commercial LLMs (since they are not open-source).

---

[1] In the causal literature this is known as the *fundamental problem of causal inference* (Holland, 1986).

The goal of this paper is to offer a theoretical foundation for the generation of counterfactuals in probabilistic LLMs that shows how the simple semantics can be taken as the general semantics for counterfactuals, whereas the complicated semantics of Chatzi et al. (2025) and Ravfogel et al. (2025) correspond to a particular deviation from this semantics that is useful for certain purposes, but not for others. The benefit of having such a foundation is that it allows a systematic exploration of different methods that each deviate from the general, simple, semantics in a manner that suits the task at hand. Below we briefly sketch several proposals for such methods, but in the current work the emphasis lies on offering a theoretical understanding of counterfactuals in LLMs, as this forms a necessary prerequisite for developing them in practice.

Informally, the general form of a counterfactual query for an LLM is of the form: "Given input $\mathbf{x}$ and output $\mathbf{y}$, what might the output $\mathbf{Y}^*$ have been if $\mathbf{x}$ were $\mathbf{x}^*$?". (One could also focus on counterfactuals involving the parameters of the LLM itself, as done by Ravfogel et al.. We intend to explore such counterfactuals in future work.) The method developed by Chatzi et al. for generating counterfactuals can be viewed as a more specific counterfactual query, of the form: "Given input $\mathbf{x}$ and output $\mathbf{y}$, what might the output $\mathbf{Y}^*$ have been if $\mathbf{x}$ were $\mathbf{x}^*$ *and* $\mathbf{Y}^*$ *were close to* $\mathbf{y}$?". In other words, their method enforces a bias towards closeness into the distribution of counterfactuals, under a specific understanding of closeness called *counterfactual stability*. This is achieved by applying the Gumbel-max trick (Gumbel, 1954; Oberst and Sontag, 2019). As illustrated in follow-up work of theirs, such counterfactuals can be used to improve the sample efficiency of comparing the quality of several different LLMs, and promises to have further applications beyond that one (Benz et al., 2025). Ravfogel et al. concurrently developed a method using the same Gumbel-based semantics as that of Chatzi et al., but focus on different applications. Crucially, both of them justify their semantics by appealing to the practical usefulness of having counterfactual stability.

Yet such a bias towards closeness is desirable for certain purposes only, but not for others. In particular, it is undesirable in the context of counterfactual explanations. Given that counterfactual explanations form one of the most prominent approaches to developing eXplainable AI (XAI), this is an important domain for future application (Wachter et al., 2017; Beckers, 2022). To compute such explanations requires the ability to generate counterfactuals with outputs that are significantly *distant* from the actual output. Roughly, for a deterministic model $f$ that generated some output $\mathbf{y}$ given factual input $\mathbf{x}$, a different input $\mathbf{x}^*$ forms a counterfactual explanation of $\mathbf{y}$ if $\mathbf{x}^*$ is very similar to $\mathbf{x}$ and yet $\mathbf{y}^* = f(\mathbf{x}^*)$ is not similar, or close, to $\mathbf{y}$. The idea is that the small differences between $\mathbf{x}$ and $\mathbf{x}^*$ allow us to identify a small set of factors that contributed substantially in generating the particular output $\mathbf{y}$. Counterfactual explanations can be generalized to probabilistic models – such as a probabilistic LLM – by considering whether the probability that the counterfactually generated $\mathbf{y}^*$ is distant to $\mathbf{y}$ is significantly large. Here the closeness property is the opposite of what we want, for it biases the distribution of probabilities of counterfactuals towards those that are close, thereby making it unnecessarily hard to find good counterfactual explanations.

There is also a growing interest in *self-generated explanations*, and here too the generation of counterfactuals has an important role to play (Agarwal et al., 2024). Concretely, we can test the faithfulness of an LLM's self-generated explanations by asking it directly to answer a counterfactual query, and then compare its answers to those we have generated ourselves by invoking the simple semantics. Dehghanighobadi et al. (2025) were the first to recently propose evaluating LLMs on self-generated counterfactual explanations, and they conclude that LLMs still perform rather poorly. Their method is restricted to deterministic LLMs, but the generation of counterfactuals for probabilistic LLMs allows their approach to be generalized to the probabilistic setting. Moving to the probabilistic setting comes with the benefit that we can instruct the LLM directly to generate counterfactuals that satisfy certain properties (such as closeness, for example), allowing one to evaluate whether it is able to generate some type of counterfactuals more faithfully than others. We can use the simple semantics as the baseline, for it is unique in being the only type of counterfactual that is available to anyone who can access the LLM, which includes the LLM itself. As a final step, the ability to generate counterfactuals could then be incorporated explicitly into reasoning models.

Relatedly, one could use the generation of counterfactuals in order to suggest alternative prompts to the user that might come closer to what the user intended, based on the fact that the actual prompt $\mathbf{x}$ results in a probability distribution reflecting very little confidence (for instance, because all outputs are assigned a very small probability) whereas some different but very similar prompt $\mathbf{x}^*$ results in assigning a large probability to only a small range of very similar outputs. For example, imagine a user asking an LLM "Is the King of France bald?". In this case, an LLM ought not to have much

confidence in any possible output. But if endowed with the ability to look for very similar prompts, it could suggest to the user whether they meant to ask about the President of France instead.

We proceed as follows. The next section describes the problem at hand and our proposed solution in general, informal, terms. Section 3 contains a brief formal introduction to both nondeterministic and deterministic causal models, which are used in Sections 4 and 5, respectively, to offer alternative representations of LLMs. Finally, we argue in favour of the former representation in Section 6, and explain how the latter can nonetheless be viewed as a useful tool for the implementation of the generation of counterfactuals that satisfy particular, application-dependent, properties.

## 2   GENERAL DESCRIPTION OF THE PROBLEM AND SOLUTION

LLMs are incredibly complicated in detail, but for starters we can describe an LLM quite simply as a family of conditional probability distributions: for each textual input there is a probability distribution over textual outputs. Whenever we prompt it with some textual input, it samples an output according to the relevant distribution. The probabilistic behavior can be made to vary by setting certain parameters $\alpha$, with one extreme being that the probabilities reduce to a function and the LLM is deterministic. An LLM does not work with text directly, but instead splits up any string into a sequence of *tokens*, and then translates each token into an integer by using a fixed dictionary. All tokens belong to a fixed LLM-specific vocabulary $V$. Concretely, an LLM $M$ is described by distributions of the form $P_M^\alpha(\mathbf{Y}|\mathbf{X})$, where both $\mathbf{X}$ and $\mathbf{Y}$ are sequences of tokens of length $< k$ for some fixed $k$. (In practice the input and output may have different lengths, but we ignore this for convenience.) The initial subsequence of $\mathbf{Y}$ is $\mathbf{X}$, i.e., the input is also returned as part of the output.

Say we prompted a given LLM $M$ with factual input $\mathbf{x}$, and it generated output $\mathbf{y}$. A counterfactual output $\mathbf{y}^*$ is an output that might have been generated by the LLM if we had asked it some different prompt $\mathbf{x}^*$ instead. The problem we (and others (Chatzi et al., 2025; Ravfogel et al., 2025)) aim to solve, is how to compute these counterfactuals for any such $\mathbf{x}^*$, and more broadly, what the meaning of such counterfactuals is in the first place. As mentioned in the introduction, there is an extremely simple solution that suggests itself: just run the LLM again on $\mathbf{x}^*$, and the output $\mathbf{y}^*$ it then generates is one instance of a counterfactual output. In order to generate more such counterfactuals, simply run the LLM again on $\mathbf{x}^*$. Formally this comes down to stating that $P^*(\mathbf{Y}^* = \mathbf{y}^*|\mathbf{Y} = \mathbf{y}, \mathbf{X} = \mathbf{x}, \mathbf{X}^* = \mathbf{x}^*) = P(\mathbf{Y} = \mathbf{y}^*|\mathbf{X} = \mathbf{x}^*)$. Here we have used the standard asterisk notation $\mathbf{V}^*$ of Balke and Pearl (1994) to distinguish counterfactual variables from factual variables $\mathbf{V}$: the latter represent events that actually take place, the former represent events that would or might have taken place if some of the actual events were different than they actually are. We use $P^*$ to denote the distribution over counterfactual variables. In other words, the *simple semantics* states that the counterfactual distribution does not depend on the factual result at all and takes on the same form as the prior, factual, distribution. As a result, the semantics of counterfactuals for the probabilistic setting is the same as that for the deterministic setting, in which the LLM is a function. The simple semantics has the major practical benefit that counterfactual queries are computed in the same way as factual queries, and thus nothing more than access to the LLM is required. Contrary to the deterministic setting, however, in the probabilistic setting there exist multiple counterfactual outputs, and therefore one needs to either run the LLM a sufficient number of times in order to estimate the distribution $P(\mathbf{Y} = \mathbf{y}^*|\mathbf{X} = \mathbf{x}^*)$, or one needs access to the LLM's source-code so that one can output the distribution after a single run. Although the former is not feasible in case the support of the distribution covers a large set of values, there exist many applications in which this is not the case, such as answers to multiple-choice questions, reasoning problems with only a limited number of sensible solutions (including queries with integer-valued answers), yes/no questions, etc. All of the case-studies evaluating the performance of different LLMs that Benz et al. (2025) use to illustrate their approach fall within this category, for example.

Obviously the practical benefit that comes with the simple semantics does not by itself form a reason to adopt it. Instead, one should adopt a semantics based on the formal properties of an LLM under its *intended interpretation* and then translate these into a theoretical framework for reasoning about counterfactuals. Fortunately such a framework is easily identified, because an LLM can be perfectly described as a causal model, and there is a rich literature on the semantics of counterfactuals within the causal modelling framework (Halpern, 2000; Pearl, 2009; Bareinboim et al., 2022; Beckers, 2025). Concretely, it is literally the case that providing an LLM with an input $\mathbf{x}$ causes it to produce

an output $\mathbf{y}$, and the mechanisms by which it does so can be described in the language of causal models. Therefore we have here a blueprint for developing a semantics of counterfactuals in LLMs, and the aim of this work is to develop it in detail. Doing so requires clarifying what exactly the causal mechanisms are by which an LLM causes its output, and then determining how these mechanisms should be represented in a causal model. We develop two approaches in detail, the first taking the intended interpretation of an LLM to be its *idealized interpretation*, the second instead taking it to be its *literal interpretation*. The Gumbel-based approach of (Chatzi et al., 2025; Ravfogel et al., 2025) is neither of these, but as mentioned earlier it can be incorporated into the idealized interpretation by viewing their method as one for generating counterfactuals that satisfy a specific property.

We briefly summarize what follows. Above we represented an LLM as just a set of conditional probabilities, but what really happens is in fact an autoregressive process, where each next token is generated during one step, and then the resulting sequence is fed as input to the next step, until the "empty" token is selected, indicating that the output has been completely generated. The crucial challenge is thus to determine how one should represent each step in which a token $T$ is generated, based on an existing sequence of tokens $\mathbf{s}$. There are two processes of interest within such a step that need to be represented, namely the generation of the distribution itself, and the sampling of a specific token according to this distribution.

Concretely, first the LLM produces a distribution $P^\alpha(T|\mathbf{s})$, where $\alpha$ are user-specified parameters, including the temperature. For convenience we leave $\alpha$ implicit, but it should be noted that different settings of $\alpha$ result in different distributions, and thus these effectively correspond to different LLMs. Other than that, for the current purposes the details of this process do not matter. Second, the LLM samples a specific token $t$ according to this distribution. The different approaches differ *only* regarding their representation of the sampling process. The literal interpretation takes into account that an LLM is running on a deterministic machine, and therefore it requires Pseudo-Random Number Generators (PRNGs) to generate behavior that *appears* to be probabilistic, but is really deterministic. Representing this formally requires using Pearl (2009)'s deterministic structural causal models, and their well-established semantics of probabilities of counterfactuals. The idealized interpretation – as its name suggests – instead represents the sampling as being ideal, meaning that it represents the token $T$ itself as a random variable distributed according to $P^\alpha(T|\mathbf{s})$. It thereby abstracts away the implementation details entirely, ignoring whatever sampling method is used in practice. Representing this formally can be done using the *nondeterministic causal models* that have recently been proposed by Beckers (2025), including a semantics for probabilities of counterfactuals. Crucially, we show that due to the very specific structure of LLMs, the idealized interpretation results in justifying the simple semantics.

The Gumbel-based approach does not clearly fit either category, but instead is motivated primarily by practical concerns. We therefore re-interpret this approach as one that does not offer a general semantics or method for the generation of counterfactuals, but rather as one that fits within the simple semantics. The result is a theoretical framework that allows for the development of methods that generate counterfactuals satisfying different properties.

## 3 Causal Models

### 3.1 Nondeterministic Causal Models

For the present purposes we can get by with using a simplified definition of nondeterministic causal models, we refer the reader to (Beckers, 2025) for the general definition and more discussion.

**Definition 1** *A nondeterministic causal model $M$ is a 3-tuple $(\mathbf{V}, \mathcal{G}, P_M)$, where $\mathbf{V}$ is a set of variables (each with their own domain), $\mathcal{G}$ is an acyclic directed graph such that there is one node for each variable in $\mathbf{V}$, and $P_M$ is a probability distribution over $\mathbf{V}$ that satisfies the Markov factorization: $P_M(\mathbf{V}) = \prod_{X \in \mathbf{V}} P_M(X|\mathbf{Pa_X})$, where $\mathbf{Pa_X}$ are the parents of $X$ in $\mathcal{G}$. If $X$ has no parents in $\mathcal{G}$, we say it is a root variable. $\mathbf{R}$ denotes the set of all root variables.*∎

For notational convenience we leave $M$ implicit whenever it is clear from the context. The *observational distribution* $P(\mathbf{V})$, represents the first layer of Pearl's Causal Hierarchy (Bareinboim et al., 2022). Given that a prompt of a user can be represented as a root variable, and given that we are not considering any interventions to the inner working of an LLM, for the present discussion we

can skip the second layer of *interventional distributions* and move on immediately to the third layer of *probabilities of counterfactuals*. The only counterfactual distribution that we need to consider in the present discussion is the probability of counterfactuals for an intervention $\mathbf{R}^* = \mathbf{r}^*$ on the root variables given actual values $\mathbf{V} = \mathbf{v}$, which is expressed as $P^*(\mathbf{V}^* = \mathbf{v}^*|\mathbf{V} = \mathbf{v}, \mathbf{R}^* = \mathbf{r}^*)$. The semantics of Beckers constructs $P^*$ by updating $P$ with the actual evidence $\mathbf{V} = \mathbf{v}$ as follows.

**Definition 2** *Given a causal model $M$ and $\mathbf{V} = \mathbf{v}$, for each $X \in \mathbf{V}$, consider the unique value $x \in \mathbf{v}$, and for each $\mathbf{pa_X}$ define $P^{\mathbf{v}}(X = x|\mathbf{pa_X}) = 1$ if $\mathbf{pa_X} \subset \mathbf{v}$, otherwise define $P^{\mathbf{v}}(X|\mathbf{pa_X}) = P(X|\mathbf{pa_X})$. Let $P^{\mathbf{v}}(\mathbf{V}) = \prod_{X \in \mathbf{V}} P^{\mathbf{v}}(X|\mathbf{Pa_X})$. We define*

$$P^*(\mathbf{V}^* = \mathbf{v}^*|\mathbf{V} = \mathbf{v}, \mathbf{R}^* = \mathbf{r}^*) = P^{\mathbf{v}}(\mathbf{V}^* = \mathbf{v}^*|\mathbf{R}^* = \mathbf{r}^*). \tag{1}$$

∎

Informally, the semantics of Beckers comes down to stating that observing some $(x, \mathbf{pa_X})$ does not affect the posterior distribution of $X$ conditional on any of the *other* – counterfactual – values $\mathbf{pa_X}' \neq \mathbf{pa_X}$, whereas the posterior distribution of $X$ conditional on $\mathbf{pa_X}$ is taken to be the standard one that assigns all probability to the actually observed value. These semantics formalize the simple idea that the only information the actual world gives us about counterfactual worlds is that the actual parents resulted in their actual children, and nothing more. Beckers introduces these semantics for models in which the probabilities reflect nondeterminism, as opposed to models in which they reflect our uncertainty regarding deterministic but unknown mechanisms. As we shall see, the latter are the subject of the well-known semantics of Pearl (2009).

We can now make precise what it means to satisfy the simple semantics. Note that this occurs precisely when $P^{\mathbf{v}}(\mathbf{V} = \mathbf{v}^*|\mathbf{R} = \mathbf{r}^*) = P(\mathbf{V} = \mathbf{v}^*|\mathbf{R} = \mathbf{r}^*)$.

**Definition 3** *We say that a causal model $M$ satisfies the* simple semantics *if for each $\mathbf{r}^*, \mathbf{v}$ with $\mathbf{r}^* \not\subseteq \mathbf{v}$, it holds that: $P_M^*(\mathbf{V}^* = \mathbf{v}^*|\mathbf{V} = \mathbf{v}, \mathbf{R}^* = \mathbf{r}^*) = P_M(\mathbf{V} = \mathbf{v}^*|\mathbf{R} = \mathbf{r}^*)$.* ∎

### 3.2 DETERMINISTIC CAUSAL MODELS

For the current purposes the following simplified definition of a deterministic causal model suffices, we refer the reader to (Pearl, 2009; Bareinboim et al., 2022) for the general definition and discussion.

**Definition 4** *A deterministic causal model $M$ is a 5-tuple $(\mathbf{V}, \mathbf{U}, \mathcal{G}, f, P_{\mathbf{U}})$, where $\mathbf{V}$ and $\mathcal{G}$ are as before, $\mathbf{U}$ is a set of variables called* exogenous *($\mathbf{V}$ are called* endogenous*), $f$ is a function $f : (\mathbf{u}, \mathbf{r}) \rightarrow \mathbf{v}$ mapping values for the exogenous and root variables to values of the endogenous variables such that $\mathbf{r} \subseteq f(\mathbf{u}, \mathbf{r})$ (i.e., $f$ is the identity function for $\mathbf{R}$), and $P_{\mathbf{U}}$ is a probability distribution over $\mathbf{U}$ satisfying positivity. A solution is a setting $(\mathbf{u}, \mathbf{r}, \mathbf{v})$ such that $\mathbf{v} = f(\mathbf{u}, \mathbf{r})$.* ∎

Instead of an unconditional observational distribution over $\mathbf{V}$, our simplified deterministic models only induce a family of conditional observational distributions of the form $P_M(\mathbf{v}|\mathbf{r}) = \sum_{\{\mathbf{u}|\mathbf{v}=f(\mathbf{u},\mathbf{r})\}} P_{\mathbf{U}}(\mathbf{u})$. Probabilities of counterfactuals – of the same restricted form as before – are now defined as: $P^*(\mathbf{v}^*|\mathbf{v}, \mathbf{r}^*) = \sum_{\{\mathbf{u}|\mathbf{v}^*=f(\mathbf{u},\mathbf{r}^*)\}} P(\mathbf{u}|\mathbf{v})$. Equivalently, this can be written as (where $\mathbf{r} \subseteq \mathbf{v}$):

$$P^*(\mathbf{v}^*|\mathbf{v}, \mathbf{r}^*) = \sum_{\{\mathbf{u}|\mathbf{v}^*=f(\mathbf{u},\mathbf{r}^*)\}} P(\mathbf{u}|\mathbf{u} \text{ is such that } \mathbf{v} = f(\mathbf{u}, \mathbf{r})). \tag{2}$$

The difference between this deterministic version of $P^*$ and the nondeterministic version (Eq. 1) is that in the former all probability is "pulled out" from the endogenous variables and is put on an additional, usually unobserved, set of exogenous variables, which then induces a probability over the endogenous variables by way of the function $f$. The semantics of Beckers and those of Pearl are thus not rivals, but are simply appropriate for different contexts. Pearl is quite explicit that he developed his semantics for deterministic contexts, where "randomness surfaces owing merely to our ignorance of the underlying boundary conditions" (Pearl, 2009, p.26).

In some extreme cases, however, a deterministic model can be given an equivalent interpretation as a nondeterministic model, due to the fact that any function can itself be interpreted as a probability distribution – taking value only in $\{0, 1\}$ – and the fact that the function $f$ does not *have* to depend on the exogenous variables. (All proofs can be found in the Appendix.)

**Theorem 1** *Given a deterministic causal model $M = (\mathbf{V}, \mathbf{U}, \mathcal{G}, f, P_{\mathbf{U}})$ such that $f$ does not depend on $\mathbf{U}$, meaning that for all $\mathbf{u}, \mathbf{u}', \mathbf{r}$: $f(\mathbf{u}', \mathbf{r}) = f(\mathbf{u}, \mathbf{r})$, there exists an equivalent nondeterministic causal model $M' = (\mathbf{V}, \mathcal{G}, P_{M'})$. Here equivalence means that $P_M^*(\mathbf{V}^*|\mathbf{V}, \mathbf{R}^*) = P_{M'}^*(\mathbf{V}^*|\mathbf{V}, \mathbf{R}^*)$ and $P_M(\mathbf{V}|\mathbf{R}) = P_{M'}(\mathbf{V}|\mathbf{R})$.*

Recall that the zero temperature – deterministic – setting of an LLM can be represented as a function $\mathbf{Y} = f(\mathbf{X})$. We can view this as a deterministic causal model of the form described in Theorem 1 by taking $\mathbf{V} = \{\mathbf{X}, \mathbf{Y}\}$, $\mathbf{U} = \emptyset$, and $\mathcal{G} = \{\mathbf{X} \rightarrow \mathbf{Y}\}$. As a consequence, the deterministic setting of an LLM can be given an accurate representation using either a deterministic causal model or a nondeterministic causal model that is empirically equivalent. This allows us to confirm the earlier claim that counterfactuals for the zero temperature case satisfy the simple semantics.

**Corollary 1** *Given a deterministic causal model $M$ that corresponds to an LLM for the zero temperature setting, $M$ satisfies the simple semantics.*

## 4 A Large Language Model as a Nondeterministic Causal Model

Now we explain how a nondeterministic LLM $M$, interpreted broadly as including the user-defined parameters, the sampling process, and any post-training adjustments, can be viewed as a nondeterministic causal model $M$. To ease notation we leave implicit that the user has specified certain parameters $\alpha$, but these could be easily integrated as variables into the causal model as well.[2]

The prompt is given by the user as a text, but it is encoded as a sequence of *tokens*, where each token corresponds to an integer. Similarly, the output is a sequence of tokens that is then decoded to give a text. An LLM has a fixed vocabulary $V$ of tokens to choose from. We assume for simplicity that both the prompt and the output (which includes the prompt as its initial subsequence) have a fixed length $k$. Given that both are sequences of tokens, we represent both as vectors $\mathbf{X} = (X_1, \dots, X_k)$ and $\mathbf{Y} = (Y_1, \dots, Y_k)$, each taking value in $V^k$. There is a special token $\emptyset$ representing the lack of text. Thus, for each prompt $\mathbf{x}$ there is some $l_{\mathbf{x}} \leq k$ such that $x_j = \emptyset$ for all $j > l_{\mathbf{x}}$. Obviously we can treat $\mathbf{x}$ as if it is a sequence of length $l_{\mathbf{x}}$ instead of $k$, and similarly for any output $\mathbf{y}$.

As the prompt is the only input to the model, $\mathbf{X}$ is the only root variable. The output is the only leaf variable (i.e., a variable with no descendants). Therefore our initial representation of the LLM from Section 1 as $P(\mathbf{Y}|\mathbf{X})$ corresponds to the probability of the leaf variable given the root variable. This means that there is no need to specify $P(\mathbf{X})$, just as was the case for deterministic causal models (Def. 4). $\mathbf{Y}$ is generated step-by-step through autoregressive token generation, meaning that we first generate a single token $t_1$ based on the input $\mathbf{x}$, then generate a second token $t_2$ based on the input $(\mathbf{x}, t_1)$, and continue this process until the empty token is generated (which becomes increasingly likely as the length of the output gets closer to $k$). Once the empty token has been generated, all subsequent tokens are empty as well. Crucially, each token generation uses the *same* family of conditional distributions $P(T|\mathbf{Z})$. Therefore we can represent the process with a causal model consisting of variables $\mathbf{V} = \{\mathbf{X}, T_1, \dots, T_k, \mathbf{Y}\}$ and the following equations and distributions:

$(T_1, \dots, T_{l_{\mathbf{x}}}) = \mathbf{X}$

for $i \in \{l_{\mathbf{x}} + 1, \dots, k\}$: $P(T_i|T_1, \dots, T_{i-1})$

$\mathbf{Y} = (T_1, \dots, T_k)$

We can now vindicate our claim that the nondetermistic causal model version of an LLM $M$ satisfies the simple semantics. The rough idea is that, if we only consider truly counterfactual prompts – meaning prompts that are not identical to the actual prompt – then the fact that the root variable $\mathbf{X}$ (or, to be precise, its representation as $(T_1, \dots, T_{l_{\mathbf{x}}})$) is a parent of all other variables implies that all of the posterior distributions are conditioned on counterfactual values and thus no updating occurs.

**Theorem 2** *Given a nondeterministic causal model $M$ that corresponds to an LLM, $M$ satisfies the simple semantics.*

---

[2]Chatzi et al. (2025) point out that when using their method whilst varying other popular sampling parameters such as top-$k$ or top-$p$, it is no longer guaranteed to result in counterfactuals that satisfy counterfactual stability. No such issues arise for the current method.

This result is a particularly strong instance of what Bareinboim et al. (2022) describe as *the collapse of the causal hierarchy*, which occurs whenever probabilities at a higher layer of the hierarchy (eg. the counterfactual layer) can be determined using probabilities at a lower layer (eg. the observational layer). Here the collapse is such that the distinction between all three layers disappears entirely, at least when restricted to conditioning only on the root variable. It is important to interpret this result correctly. What it shows, is that – due to the very unique structure of LLMs – counterfactual distributions and observational distributions have the same *extension*, meaning that they take on the same form. It does not show that they have the same *intension*, meaning that they do not share the same interpretation. Counterfactuals express what might have happened if things had been different, and observations express what actually happened. That the one takes on the same form as the other is a discovery that we can use to our advantage, and that is the main take-away of this work.

## 5  A LARGE LANGUAGE MODEL AS A DETERMINISTIC CAUSAL MODEL

The foregoing results crucially depend on modelling an LLM as a *nondeterministic* causal model. We now consider what things look like if we were to model an LLM as a deterministic causal model. Except for some minor technical differences, our model closely follows that of Chatzi et al. (2025).

We need to find a function $f$ and exogenous variabes $\mathbf{U}$ such that $(\mathbf{Y}, T_k, \ldots, T_1, \mathbf{X}) = f(U_1, \ldots, U_k, \mathbf{X})$. This means we need to "pull out" the probability from each $P(T_i|T_1, \ldots, T_{i-1})$ for $i > l_\mathbf{x}$ and put it onto some new exogenous variable $U_i$, so that we get an equation of the form $T_i = g(U_i, T_1, \ldots, T_{i-1})$ for some function $g$ and a probability distribution $P(U_i)$ satisfying $P(t_i|t_1, \ldots, t_{i-1}) = \sum_{\{u_i|t_i=g(u_i,t_1,\ldots,t_{i-1})\}} P(u_i)$. Take note that, importantly, in general there will be many choices for $U_i$, $g$, and $P(U_i)$, that satisfy this constraint. As each $T_i$ is drawn independently from the identical distribution $P(T|\mathbf{Z})$, the new variables $U_i$ will be mutually independent and identically distributed. As a result, we obtain a deterministic model with exogenous variables $\mathbf{U} = (U_1, \ldots, U_k)$, distribution $P_\mathbf{U}(\mathbf{U}) = \prod_{i \in \{1,\ldots,k\}} P(U_i)$. We can now take $f$ to be decomposed into $(T_1, \ldots, T_{l_\mathbf{x}}) = \mathbf{X}$ and the recursive application of $T_i = g(U_i, T_1, \ldots, T_{i-1})$ for $i > l_\mathbf{x}$.

As noted, this construction allows for many different choices of $U_i$, $g$, and $P(U_i)$ that each result in a deterministic causal model $M$ such that its observational distribution is identical to the distribution $P(\mathbf{Y}|\mathbf{X})$ of a given LLM $M$. These choices can result in wildly diverging values for probabilities of counterfactuals, which is why their partial identifiability is one of the major research challenges in the literature on causal models (Pearl, 2009; Zhang et al., 2022). Here is a simple example.

**Example 1** *We want to construct a causal model $M$ of some very simple LLM with binary endogenous variables $X$ and $Y$, a graph $X \to Y$, and some values $0 < p < q < 1$ such that $P(Y = 1|X = 1) = p$ and $P(Y = 1|X = 0) = q$. If we assume that the causal model is deterministic, then we have to pull out the probabilities by adding exogenous variables. In this simple case the standard canonical choice – that has been proven in general by Zhang et al. (2022) to be sufficiently expressive to cover all possible underlying models – is to add a four-valued $U$ so that $Y$ is determined by $X$ and $U$ as follows: $Y = X$ if $U = 0$, $Y = \neg X$ if $U = 1$, $Y = 0$ if $U = 2$, and $Y = 1$ if $U = 3$. Observe that any choice of $P(U)$ such that $p = P(U = 0) + P(U = 3)$ and $q = P(U = 1) + P(U = 3)$ satisfies the requirement that $P(Y = 1|X = 1) = p$ and $P(Y = 1|X = 0) = q$. Choosing, for example $P(U = 3) = 0$, $P(U = 0) = p$, and $P(U = 1) = q$, gives $P^*(Y^* = 0|Y = 1, X = 1, X^* = 0) = 1$. Yet choosing $P(U = 3) = p$, $P(U = 0) = 0$, and $P(U = 1) = q - p$, gives $P^*(Y^* = 0|Y = 1, X = 1, X^* = 0) = 0$.[3] In other words, the probabilities of counterfactuals for $Y^*$ are entirely unbounded.*

*On the other hand, if we assume that the causal model is nondeterministic, then by Theorem 2 it satisfies the simple semantics, and thus $P^*(y^*|x, y, x^*) = P(y^*|x^*)$ for all values $y^*, y, x \neq x^*$. In other words, the probabilities of counterfactuals for $Y^*$ are point-identified.* ∎

To sum up, *if* we only have access to the LLM's overall behavior in the form of a distribution $P(\mathbf{Y}|\mathbf{X})$, and *if* we model the LLM as a deterministic causal model, we are unable to compute probabilities of counterfactuals, and a fortiori we are unable to faithfully generate counterfactuals.

---

[3]As these choices violate positivity, they are not allowed by Definition 4. This technicality can be overcome by working with two different three-valued exogenous variables $U_1$ and $U_2$ for each of the two choices instead.

A natural suggestion is to consider whether also having access to some of the implementation details of an LLM could offer sufficient information to identify unique choices for $U_i$, $g$, and $P(U_i)$ that accurately represent the inner workings of the LLM, in which case probabilities of counterfactuals become identifiable after all. Furthermore, if we could in addition also get access to the values of $\mathbf{U}$ for each run, then the computation of counterfactuals becomes entirely deterministic, for it follows directly from (2) that $P^*(\mathbf{y}^*|\mathbf{y}, \mathbf{x}, \mathbf{u}, \mathbf{x}^*)$ takes value only in $\{0, 1\}$. Let us unpack this suggestion.

Each iteration of the autoregressive token generation process consists of two distinct processes: the first process takes a sequence of tokens $\mathbf{s_{i-1}}$ and outputs a distribution $P(T_i|\mathbf{s_{i-1}})$ over the LLM's fixed vocabulary $V$, the second process then samples a token $t_i$ from $V$ according to that distribution. The first process is deterministic and therefore its implementation details are of no concern to us here. The second process, however, requires the implementation of a *sampling method*, and that lies at the heart of the issue under investigation. For example, if the token 'love' has probability $0.01$, then when running this process $1,000,000$ times such a method should return 'love' approximately $10,000$ times. The ideal sampling method would be to have direct access to a random variable $T_i$ that is distributed as $P(T_i|\mathbf{s_{i-1}})$. Imagine, for example, that the distribution is the uniform distribution over a binary outcome, then the ideal method would be to simply flip a perfectly fair coin.

As LLMs are implemented on computers, and as almost all computers are deterministic machines, they do not have access to such truly random variables. Therefore in practice the ideal sampling method is approximated by a sampling method that uses a PRNG (Pseudo-Random Number Generator) instead. When called a PRNG generates *what appears to be* a random number in the real interval $[0, 1)$. For example, say the PRNG returns $p$ (which we could represent in the causal model as $U_i = p$), then we can use $p$ to sample a token $T_i$ by applying inverse transform sampling to $P(T_i|\mathbf{s_{i-1}})$: given some fixed ordering $t^1, \dots, t^{|V|}$ of all tokens in $V$, we return the token with the lowest index $n$ such that the probability conditional on $\mathbf{s_{i-1}}$ of obtaining some token with index $m \le n$ is greater than $p$. As required, this translates into a deterministic equation for $T_i$ of the form $T_i = g(U_i, \mathbf{S_{i-1}})$, where $g(U_i, \mathbf{S_{i-1}}) = t^n$ for the unique value $n$ such that $P(T \in \{t^0, \dots, t^n\}|\mathbf{s_{i-1}}) > U_i$. A PRNG is a deterministic function that *appears* to behave non-deterministically by making use of some highly contingent detail that for most intents and purposes can be treated *as if* it is random, such as the time of execution, or the CPU time, or some seed that is hard-baked into the source code, or a seed that is determined by some other program, etc. This means that faithfully modeling the particular sampling method used by an LLM as a deterministic causal model requires spelling out intricate implementation details of the particular PRNG and the particular method used for setting the seed, resulting in some function $U_i = s(\beta)$, where $U_i$ is the pseudo-random number taking value in $[0, 1)$ as before, $\beta$ are the parameters that aim to mimic a random source of variation, and $s$ represents the implementation details that allow the latter to be transformed into the former. Unfortunately, as pointed out by Chatzi et al. (2025), an LLM is stateless, meaning that it does not have an internal memory that keeps track of all the steps in its computation, and therefore in practice the method here outlined is not possible.

## 6 THE INTENDED INTERPRETATION OF LLMS

We now have two approaches to interpreting an LLM, the idealized interpretation from Section 4 that idealizes the sampling process as being truly random, and the literal interpretation just discussed. The first results in the simple semantics and is therefore practically trivial to implement. The second results in a semantics according to which the fine-grained details of the sampling process are crucial and is therefore practically difficult to implement. So from a practical perspective, the idealized interpretation is to be preferred. We contend that the idealized interpretation is also to be preferred from a theoretical perspective, for it accords much better with the intended interpretation of an LLM.

To make this clear, consider for example an LLM implemented with a sampling process that uses the time of execution as a seed. Imagine we ran the LLM on Tuesday on prompt $\mathbf{x}$ and observed output $\mathbf{y}$. Under the literal interpretation, the answer to the question "What would have been the most likely output if the prompt were $\mathbf{x}^*$ instead of $\mathbf{x}$?" would be entirely different from the answer to the question "What would have been the most likely output if the prompt were $\mathbf{x}^*$ instead of $\mathbf{x}$ *and it was Wednesday instead of Tuesday*?". More generally, no matter what the actual details of the specific sampling process are, one can come up with entirely irrelevant changes – such as the time of execution, the CPU time, the state of the random number generator, etc. – so that the distribution

of counterfactuals takes on an entirely different form. As a result, the usefulness of a literal interpretation is restricted entirely to that exact specific implementation. Such low-level implementation sensitivity applies also to literal interpretations of probabilistic programs, and that is precisely why it is standard to interpret probabilistic programs under a *denotational semantics* instead (Kozen, 1981). Notably, just as is the case with nondeterministic causal models, under the denotational semantics probability distributions are represented as primitives of the program and they can be sampled directly, abstracting away all details of how the sampling process is actually implemented. As we have shown above, within the context of LLMs this idealized, intended, interpretation results in the simple semantics for counterfactuals.

Chatzi et al. (2025) and Ravfogel et al. (2025) implicitly adopt an interpretation of LLMs that sits somewhere in between both. It also represents the sampling process using Pearl's deterministic framework, and therefore uses his semantics as the basis of their method. They overcome the practical problem facing the literal interpretation by replacing the source code of the sampling process by an alternative sampling method that is explicitly designed so that it satisfies two properties: the details necessary for the purposes of generating counterfactuals are explicity stored, and the counterfactuals so generated are intentionally biased towards those that are close. (Concretely, the sampling method implements a so-called Gumbel-Max deterministic causal model, and it stores the values of $\mathbf{U}$ of each run so that counterfactual outputs can be generated deterministically (Oberst and Sontag, 2019).) Lastly, to make their approach feasible, they add one layer of idealization between the PRNG and the variables they use to compute counterfactuals. Neither Chatzi et al. nor Ravfogel et al. offer a theoretical justification of their approach, nor do they present it as a general semantics for counterfactual generation in LLMs. Rather, they justify their method based on the fact that it can actually be implemented, and on the fact that close counterfactuals are useful for certain purposes. In fact, they both acknowledge that their method is not appropriate for all situations and suggest both exploring alternative implementations as well as different properties than closeness.

In light of all this, their approach can be interpreted as one that offers a method that enforces a particular bias into the distribution of counterfactuals, whose unbiased distribution is captured by the simple semantics. Formally, this can be expressed in our framework as follows.

**Definition 5** *Given a nondeterministic causal model M, we define the* counterfactually stable distribution $P_{st}^*$ *as* $P_{st}^*(\mathbf{V}^* = \mathbf{v}^* | \mathbf{V} = \mathbf{v}, \mathbf{R}^* = \mathbf{r}^*) = \prod_{\{Y \in \mathbf{V} \setminus \mathbf{R}\}} P(y^* | \mathbf{pa_Y}^*, y^* \notin C_Y) 1_{\mathbf{r}^* \subseteq \mathbf{v}^*}$ *where* $C_Y$ *is given as* $C_Y = \{y' \neq y \text{ s.t. } \frac{P(y | \mathbf{pa_Y}^*)}{P(y | \mathbf{pa_Y})} \geq \frac{P(y' | \mathbf{pa_Y}^*)}{P(y' | \mathbf{pa_Y})}\}$, *with* $(\mathbf{pa_Y}, y) \subseteq \mathbf{v}$. ∎

If our aim is to generate counterfactuals whose distribution satisfies some other property, then we should consider a different implementation, corresponding to a different deterministic causal model than the Gumbel-Max one. For example, one could consider an alternative definition of closeness that uses a metric over the output space and only considers outputs that are at most $\epsilon$ removed from the actual output. Analogously, one could look for counterfactuals that are at least $\epsilon$ removed: generating such counterfactuals would be useful for exploring how different the range of outputs might have been even if we only changed the prompt slightly. Crucially, these different implementations can all be viewed as implementations of the same LLM, that is captured by a single nondeterministic causal model, so that we can systematically express different types of counterfactuals as introducing different types of biases into the unbiased distribution that follows the simple semantics.

## 7 CONCLUSION

We have developed a formal framework for counterfactuals in LLMs by representing them as nondeterministic causal models, proving that it results in the simple semantics according to which probabilities of counterfactuals take on the same form as observational probabilities. As a result, the generation of counterfactuals for probabilistic LLMs can proceed identically to that of deterministic LLMs, and is available to all. We compared our approach to the Gumbel-based approach that represents LLMs as deterministic causal models, and requires access to an LLM's source-code to generate counterfactuals. Rather than rival approaches to the same problem, we showed that they can be viewed as complementary: the Gumbel-based method offers an implementation for the generation of counterfactuals that satisfy a particular property that is desirable for some applications. Our theoretical analysis forms the basis upon which other properties that are useful for other applications, and the practical methods for implementing them, can now be developed.

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

APPENDIX: PROOF OF THEOREMS

Beckers (2025) shows that Equation (1) can be written as follows, which will prove useful in establishing the results below:

$$P^*(\mathbf{V}^* = \mathbf{v}^* | \mathbf{V} = \mathbf{v}, \mathbf{R}^* = \mathbf{r}^*) = \begin{cases} 0 \text{ if } \mathbf{r}^* \not\subseteq \mathbf{v}^* \\ 0 \text{ if } \emptyset \neq \{Y \in \mathbf{V} \setminus \mathbf{R} | \mathbf{pa_Y} = \mathbf{pa_Y}^* \text{ and } \mathbf{y}^* \neq \mathbf{y}\} \\ 1 \text{ if } \emptyset = \{Y \in \mathbf{V} \setminus \mathbf{R} | \mathbf{pa_Y} \neq \mathbf{pa_Y}^*\} \\ \prod_{\{Y \in \mathbf{V} \setminus \mathbf{R} | \mathbf{pa_Y} \neq \mathbf{pa_Y}^*\}} P(y^* | \mathbf{pa_Y}^*) \text{ otherwise.} \end{cases} \quad (3)$$

Roughly, the four cases of (3) correspond to the following properties:

1. The counterfactual antecedent has to hold, as is standard.

2. Any counterfactual world in which some child variable $Y$ obtains a non-actual value despite all of its parents taking on their actual values, is excluded. The motivation for this is that the actual world offers information regarding the behavior of the nondeterministic mechanism given the actual parent values, and this behavior is identical in any world that is most similar to the actual given the counterfactual antecedent. This is a consequence of adopting the general idea of a closest possible world semantics that was developed by Lewis (1973) and taken over by Pearl (2009).

3. This case almost comes down to stating that if the counterfactual antecedent is consistent with the actual values, then the only possible world is the actual world. To see why, note first that the condition says there is no variable that is both a parent and takes on a non-actual value, and second note that the leave variables that do have parents have to also take on actual values by the second case. The only kind of variables that do not satisfy either characterization are root variables that do not have any children, and thus only those form an exception to the initial statement.

4. The fourth case states roughly that for all the counterfactual worlds failing to satisfy any of the earlier cases, we use the prior distribution for all variables that have parents with non-actual values. (Note that the variables which have parents with actual values take on their actual values, by the second case.)

**Theorem 1** *Given a deterministic causal model $M = (\mathbf{V}, \mathbf{U}, \mathcal{G}, f, P_{\mathbf{U}})$ such that $f$ does not depend on $\mathbf{U}$, meaning that for all $\mathbf{u}, \mathbf{u}', \mathbf{r}$: $f(\mathbf{u}', \mathbf{r}) = f(\mathbf{u}, \mathbf{r})$, there exists an equivalent nondeterministic causal model $M' = (\mathbf{V}, \mathcal{G}, P_{M'})$. Here equivalence means that $P_M^*(\mathbf{V}^* | \mathbf{V}, \mathbf{R}^*) = P_{M'}^*(\mathbf{V}^* | \mathbf{V}, \mathbf{R}^*)$ and $P_M(\mathbf{V} | \mathbf{R}) = P_{M'}(\mathbf{V} | \mathbf{R})$.*

**Proof:** Assume $M = (\mathbf{V}, \mathbf{U}, \mathcal{G}, f, P_{\mathbf{U}})$ is a deterministic causal model such that $f$ does not depend on $\mathbf{U}$. Define the nondeterministic model $M'$ as follows: $\mathbf{V}' = \mathbf{V}$, $\mathcal{G}' = \cup_{\{R \in \mathbf{R}, Y \in \mathbf{V} \setminus \mathbf{R}\}} \{R \rightarrow Y\}$. Further, choose a constant value $\mathbf{u}'$ at random, and define $P_{M'}$ as the joint distribution over the following factorization. There is a uniform distribution $P(R)$ for each $R \in \mathbf{R}$, and for each $Y \in \mathbf{V} \setminus \mathbf{R}$, $P(Y | \mathbf{R})$ is the projection of $f(\mathbf{u}', \mathbf{R})$ onto $Y$.

It follows directly that $P_M(\mathbf{V} | \mathbf{R}) = 1_{\{\mathbf{V} = f(\mathbf{u}', \mathbf{R})\}} = P_{M'}(\mathbf{V} | \mathbf{R})$, where $1_{condition}$ is the indicator function returning 1 if the condition holds and 0 if it does not. Given $\mathbf{v}, \mathbf{r}^*$, remains to be shown that $P_M^*(\mathbf{V}^* | \mathbf{v}, \mathbf{r}^*) = P_{M'}^*(\mathbf{V}^* | \mathbf{v}, \mathbf{r}^*)$.

We have that $P_M^*(\mathbf{V}^* | \mathbf{v}, \mathbf{r}^*) = \sum_{\{\mathbf{u} | \mathbf{V}^* = f(\mathbf{u}, \mathbf{r}^*)\}} P(\mathbf{u} | \mathbf{v}) = 1_{\{\mathbf{V}^* = f(\mathbf{u}', \mathbf{r}^*)\}}$. (Note that, combined with the above, this means that $M$ satisfies the simple semantics.)

First we consider $P_{M'}^*(\mathbf{v}^* | \mathbf{v}, \mathbf{r}^*)$ for the unique value $\mathbf{v}^* = f(\mathbf{u}', \mathbf{r}^*)$. Per definition of a deterministic model, $f$ restricted to $\mathbf{R}$ is the identity function, and thus the first case of (3) does not apply. Consider first the scenario such that $\mathbf{r} = \mathbf{r}^*$, where $\mathbf{r} \subseteq \mathbf{v}$. This means that $\mathbf{v} = \mathbf{v}^*$, and thus the second case of (3) does not apply. The third case does apply, since $\mathbf{R} = \mathbf{Pa_Y}$ for all $Y \in \mathbf{V} \setminus \mathbf{R}$, and therefore $P_{M'}^*(\mathbf{v}^* | \mathbf{v}, \mathbf{r}^*) = 1 = 1_{\{\mathbf{v}^* = f(\mathbf{u}', \mathbf{r}^*)\}}$, as required. Consider second the scenario such that $\mathbf{r} \neq \mathbf{r}^*$. Then by the same reasoning only the fourth case of (3) applies, and thus $P_{M'}^*(\mathbf{v}^* | \mathbf{v}, \mathbf{r}^*) = P_{M'}(\mathbf{v}^* | \mathbf{r}^*) = 1_{\{\mathbf{v}^* = f(\mathbf{u}', \mathbf{r}^*)\}}$, as required. ∎

**Corollary 1** *Given a deterministic causal model $M$ that corresponds to an LLM for the zero temperature setting, $M$ satisfies the simple semantics.*

**Proof:** This is a direct consequence of Theorems 1 and 2. ∎

**Theorem 2** *Given a nondeterministic causal model $M$ that corresponds to an LLM, $M$ satisfies the simple semantics.*

**Proof:** Say $M = (\mathbf{V}, \mathcal{G}, P_M)$ is a nondeterministic causal model $M$ that corresponds to an LLM. Note that $\mathbf{R} = \mathbf{X}$, and thus we only need to consider counterfactuals of the form "If $\mathbf{X}^*$ were $\mathbf{x}^*$".

We have that $\mathbf{V} = \{\mathbf{X}, T_1, \ldots, T_k, \mathbf{Y}\}$, and given some $\mathbf{v} = (t_k, \ldots, t_1, \mathbf{x}, \mathbf{y})$ and $\mathbf{x}^*$ with $\mathbf{x}^* \neq \mathbf{x}$, we need to show that $P^{(t_k, \ldots, t_1, \mathbf{x}, \mathbf{y})}(\mathbf{V} = \mathbf{v}^* | \mathbf{X} = \mathbf{x}^*) = P(\mathbf{V} = \mathbf{v}^* | \mathbf{X} = \mathbf{x}^*)$. Given the equations $(T_1, \ldots, T_{l_\mathbf{x}}) = \mathbf{X}$ and $\mathbf{Y} = (T_1, \ldots, T_k)$, it suffices to show that for each valuation $(t^*_{l_\mathbf{x}+1}, \ldots, t^*_k)$:

$$P^{(t_k, \ldots, t_1, \mathbf{x}, \mathbf{y})}(t^*_{l_\mathbf{x}+1}, \ldots, t^*_k | t^*_1, \ldots, t^*_{l_\mathbf{x}}) = P(t^*_{l_\mathbf{x}+1}, \ldots, t^*_k | t^*_1, \ldots, t^*_{l_\mathbf{x}}).$$

First, note that per the Markov factorization, $P(t^*_{l_\mathbf{x}+1}, \ldots, t^*_k | t^*_1, \ldots, t^*_{l_\mathbf{x}}) = \prod_{i \in \{k, \ldots, l_\mathbf{x}+1\}} P(t^*_i | t^*_{i-1}, \ldots, t^*_1)$.

Second, by the definition of $P^\mathbf{v}$, we have that $P^{(t_k, \ldots, t_1, \mathbf{x}, \mathbf{y})}(t^*_{l_\mathbf{x}+1}, \ldots, t^*_k | t^*_1, \ldots, t^*_{l_\mathbf{x}}) = \prod_{i \in \{k, \ldots, l_\mathbf{x}+1\}} P^{(t_k, \ldots, t_1, \mathbf{x}, \mathbf{y})}(t^*_i | t^*_{i-1}, \ldots, t^*_1)$.

Third, note that for each $i \in \{k, \ldots, l_\mathbf{x}+1\}$, the parents of $T_i$ are a superset of $\{T_1, \ldots, T_{l_\mathbf{x}}\}$. Given that $\mathbf{x}^* \neq \mathbf{x}$, meaning that $(t^*_1, \ldots, t^*_{l_\mathbf{x}}) \neq (t_1, \ldots, t_{l_\mathbf{x}})$, by definition of $P^\mathbf{v}$ we have that $P(t^*_i | t^*_{i-1}, \ldots, t^*_1) = P^{(t_k, \ldots, t_1, \mathbf{x}, \mathbf{y})}(t^*_i | t^*_{i-1}, \ldots, t^*_1)$, and thus the result follows. ∎

