# OpenReview forum: "Large Language Models as Nondeterministic Causal Models"
_ICLR.cc/2026/Conference — ICLR 2026 Conference Withdrawn Submission_

### Official Review · Reviewer_oUCw · 2025-10-21

**Soundness:** 3
**Presentation:** 3
**Contribution:** 3
**Rating:** 6
**Confidence:** 3

**Summary:**

The paper addresses the question of what is a counterfactual of a string generated by a language model and discusses the exact semantics under multiple interpretations of a language model as a causal model. In particular, the paper proposes framing language models as nondeterministic causal models. This interpretation allows for a simple definition of a string counterfactual as simply another sample from the same language model (given the same prompt). This is in contrast with the interpretation as a deterministic language model, which is closer to what recent work has framed language models as. In particular, the proposed counterfactual sampling is simpler than previous proposed techniques that require access to the full model implementation, make assumptions about the sampling process, and rely on changing model weights. The paper also discusses when each of the interpretations is more useful.

**Strengths:**

- The paper addresses a relevant and interesting question.
	- The problem and the setup are well-motivated and explained; for example, it makes it very clear where generating counterfactuals is useful
	- It also critically evaluates recent work on framing LMs as causal models
- The paper discusses the problem very thoroughly
	- I feel like it represents a good reference on the topic of counterfactual generation in LMs
- Provides a new theoretically-backed way of sampling counterfactuals that’s actually substantially easier and more easily applicable than existing methods
- The discussion of deterministic LMs is helpful and illustrative

**Weaknesses:**

- I found the paper quite dense and theoretical on the first read. I’m not sure how to address that, though; the examples and connection to existing techniques already help.
- I understand that the paper is theoretical and it’s primary goal is to formalize what a counterfactual could mean. However, I imagine a practical demonstration of what counterfactuals are and how they differ under different formalisms would be useful.
- Nit-picky: I found the notation slightly confusing and I believe the exposition could benefit from introducing the notation explicitly; e.g., what are upper and lower-case variable names and how to interpret statements such as subset
	- Maybe a more thorough introduction of nondeterministic causal models in the appendix (or the extra page) could be useful

**Questions:**

- I’m slightly confused by Definition 2; isn’t $\mathbf{pa_X} \subset \mathbf{v}$ always true?
- Do the frameworks need to change substantially if the assumption of finite length ($k$) is lifted?
- It seems to me that unlike Chatzi et al.’s and this paper’s frameworks, Ravfogel et al.’s framework works with interventions. Is that a correct reading?
- I understand the nondeterministic causal model is a properly defined concept, but how practically useful is it to say that observing some outcome tells you nothing about other possible counterfactual distributions?

---

> ### Author Response · Authors · 2025-11-15
>
> We thank the reviewer for their positive evaluation.
>
> On the point of clarifying the different views on counterfactuals using examples, we do offer such an example, see Example 1. Furthermore, counterfactual outputs under the simple semantics look identical to observational outputs, so we saw no point in empirically illustrating what this looks like, as it looks like the vanilla behavior of the LLM.
>
> We take the point on notation, and will be more explicit in clarifying it earlier one. Adding a more in-depth discussion of nondeterministic causal models to the appendix is definitely a worthwhile suggestion that we will take up.
>
> __Def 2:__ This confusion is probably due to the confusion regarding our notation. Lower-case letters refers to values of the variables. So indeed, ${\bf Pa_X} \subset {\bf V}$ is always true. But ${\bf pa_X} \subset {\bf v}$ means that ${\bf Pa_X}={\bf pa_X}$ are the actual values occurring in the actual state ${\bf V} = {\bf v}$.
>
> __Finite length:__ we are not sure, to be honest. Ravfogel et al. (2025) do generalize their approach to the infinite setting, so this suggests that additional complications do arise. However, note that it is quite unorthodox to expand causal models to include an infinite number of variables, as this has been suggested only recently in Peters & Halpern (2021) “Causal modeling with infinitely many variables”.
>
> The reviewer is correct that Ravfogel et al. discuss interventions on the LLM in addition to mere counterfactual prompts, whereas our paper and that of Chatzi et al. do not. We are currently exploring how to expand our approach to such interventions, as that is definitely of great practical interest.
>
> The practical use should be guided by the concrete goal that one has in mind. For example, as we state in the paper, if one is using counterfactual generation to increase the sample efficiency of comparing the quality of different LLMs, such as in (Benz et al.), then the method of Chatzi et al. is entirely appropriate. On the other hand, if one is interested in using counterfactuals for generating counterfactual explanations, one ought to use either our semantics as is, or construct a novel method that allows for generating counterfactuals that are distant from the actual one. As far as we know, no such method currently exists. We believe that work on such alternative methods was hindered by the lack of a general theoretical framework, and we are thus confident that this will change upon publication of our paper.

---

> > ### Comment · Reviewer_oUCw · 2025-11-24
> >
> > Thank you for the clarifications! I was indeed confused about the notation---I did not realize the difference between $\textbf{Pa}$ and $\textbf{pa}$.
> > I believe that including more background in the appendix would definitely be useful. I maintain my positive evaluation of the paper and hope it is accepted.

---

### Official Review · Reviewer_EKa7 · 2025-10-28

**Soundness:** 1
**Presentation:** 2
**Contribution:** 1
**Rating:** 2
**Confidence:** 3

**Summary:**

The work proposes that one can sample counterfactuals from a language model by simply querying a model again with a different (counterfactual) prompt. This is termed the **simple semantics**.

Compare this to prior work, where the (afaik) more standard approach of maintaining exogenous noise in a structural causal model is done. The work says this is in some way more complicated.

**Strengths:**

Counterfactual sampling from LLMs is, at least to me, a non-intuitive field of research. Providing clarity and simplification to the field and shared understanding of how to do counterfactual sampling is an important goal.

**Weaknesses:**

The four-page introduction chapters left me somewhat confused. I think the paper can be made more accessible by being more precise earlier on. For instance, the core object of study, an LLM, is not specified as is done more traditionally, i.e., as a distribution over finite strings in some language, but as some sort of conditional. Phrases like "LLMs are incredibly complicated in detail" don't say much; there are good standard definitions used in the cited works.

More seriously, I feel it is a bit of a strawman argument that there is something overly complicated going on in the prior work that uses the Gumbel approach. It only uses the output logits of the LLMs. The talk of changing source code or having access to the internals of LLMs doesn't exactly paint the full picture; it is a small change in sampling given the output logits, which are sometimes available for closed-source models (as opposed to the normalized probabilities). The problem is thus overstated a few times. The central problem is that randomness needs to be **coupled** to talk about counterfactuals; just running vanilla inference can not do this unless the randomness is controlled for somehow. I am also not sure that the prior works (Chatzi and Ravfogel) explicitly say the "simple semantics" is unsound -- but at least they would say it answers a different question. Stating that "Neither Chatzi et al. nor Ravfogel [spelling mistake in paper] offer a theoretical justification of their approach." is not right, it directly follows the standard SCM definition of what it means to be a counterfactual. And talk about changing the LLM in the abstract misses the point that Gumbel sampling is the same as categorical sampling, as is covered well in the related work.

Second, the core point in Theorem 2 on **the collapse of the causal hierarchy** does not feel like the full picture for what we mean by a standard counterfactual. The prior work can ask what the noise is that produces an outcome, and then apply it to a different input. Theorem 2 does not do anything of that sort; it only relies on the input X. Doesn't this just miss the point of what it means to have a pointwise counterfactual rather than a population sample?

Finally, there is no empirical demonstration, despite claims about the approach being directly applicable to any LLM. This would have been good to see.

The paper raises an interesting perspective, but it currently overstates the complexity of prior work and frames a narrower semantics than standard abduction-based counterfactuals. With a clearer introduction (and a more structured first half), a neutral comparison to Gumbel coupling, a scoped claim for Theorem 2, and a small empirical demonstration, the contribution would be stronger. But the critique of pointwise counterfactuals seems, despite the philosophical discussion, unwarranted, and thus I can not recommend that the paper be accepted.


Smaller notes:
* Temperature 0 as determinism doesn't hold for all sampling methods
* Black boxes at the end of definitions.
* missing definition of u,v,r in definition 4.
* references used without links (cref)
* again, the prior work comparison needs to be done more fairly and transparently

**Questions:**

Can you comment on the point above about the collapse of the causal hierarchy?

---

> ### Author Response · Authors · 2025-11-15
>
> We thank the reviewer for their interaction with our work.
>
> The main objection the reviewer seems to have, is that we overstate the complications involved in the Gumbel approach, the reason being that the change in sampling is relatively easy to implement. We accept their point that fully open source models are indeed not required, since the output logits are indeed sufficient, so we will correct this. However, the criticism we have of the Gumbel approach is not primarily a criticism of its practical feasibility, but rather a criticism of its failure to offer a general semantics of counterfactual generation, and thus it being entirely ad hoc. We fear that without offering a more theoretical and more general framework as the one that we develop, future work on generating counterfactuals, undoubtedly involving many other methods than the Gumbel one, will be conceptually confused. Instead, we aim to offer a framework that clarifies how existing and future methods relate to each other.
>
> So to be clear, we agree with the reviewer that all that needs to be changed for the Gumbel approach is the sampling method. We entirely disagree that this is a small change: quite the contrary, under the deterministic SCM view, the sampling method _entirely_ determines the counterfactual distribution. (The only requirement is that it induces the correct observational distribution.) See Example 1, where different choices result in the probability changing from 0 to 1. Surely there cannot be a bigger difference than that? What this means, is that without any theoretical basis for understanding counterfactuals of LLMs, it is open to anyone to choose a completely different sampling method than the one using the Gumbel trick, and they will generate entirely different counterfactuals, even though the observed behavior of the LLM is _identical_. This is the point we make regarding the extreme sensitivity under the literal interpretation of an LLM, see section 6.
>
> Furthermore, the reviewer states that “The central problem is that randomness needs to be coupled to talk about counterfactuals;”. This claim is true only under the Pearlian semantics, according to which all randomness is due to unobserved latent variables, and it is precisely the point of our paper that this semantics is inappropriate when modelling probabilistic LLMs (just as it is inappropriate in the context of probabilistic programs, see line 435 on the denotational semantics). The reviewer claims that the other authors do offer a theoretical justification, because they use the “standard SCM definition of what it means to be a counterfactual”. That is not a theoretical justification though, for it avoids addressing the fundamental question: why should we represent an LLM as an SCM to begin with? If one believes that the Pearl-style SCM semantics are the only possible semantics for counterfactuals that exist, then this question would not arise. But that belief is nothing but a dogma that is oblivious to the rich history on the semantics of counterfactuals, going back fifty years to the pivotal work of Lewis and Stalnaker.
>
> Regarding Theorem 2 missing the point, we can only re-iterate that the point of the paper is precisely to criticize the unreflective adoption of the SCM-based Pearlian based semantics. We ask the reviewer to consider the following counterfactual. Imagine they had never heard of SCMs, nor of causal models even, and had no prior commitments to any specific theoretical semantics for counterfactual. In other words, imagine one is not a researcher in AI. Then consider the reasoning we present in the first paragraph. Does it not make perfect sense? Perhaps we should add a user-study to our paper, for we are confident that people who have not been exposed to Pearl-style counterfactuals would agree that it does. (And reviewer P1G1 seems to agree as well, as they claim that “the main contribution formalizes an intuition already shared by many practitioners.”)
>
> We thank the reviewer for the smaller notes, and will adapt the paper accordingly.

---

> > ### Comment · Reviewer_EKa7 · 2025-11-25
> >
> > Thanks for the response and pushing back. I have given it further thought and see your point. Given the existing literature, and that while these counterfactuals are possible, I am inclined to agree that they are artifacts of the sampling method. And it makes an important point about the counterfactual mechanisms. While the Gumbel and CDF approaches can potentially give useful counterfactuals in some settings, they are arbitrary and should be recognized as such.

---

### Official Review · Reviewer_x8hu · 2025-10-30

**Soundness:** 2
**Presentation:** 1
**Contribution:** 3
**Rating:** 4
**Confidence:** 3

**Summary:**

This paper provides a new theoretical foundation for generating counterfactuals in Large Language Models (LLMs). Counterfactuals answer the question: "What would the output ($Y^\*$) have been if the input ($X$) were different ($X^*$)?"

The authors model the LLM's behavior using a nondeterministic causal model, which contrasts with prior work that used deterministic models with complex, implementation-dependent semantics (like Gumbel-based methods) that enforce an arbitrary "closeness" between the factual and counterfactual outputs.

The paper proves that, given the LLM's autoregressive structure, the correct causal interpretation justifies the "simple semantics": the counterfactual distribution $P(Y^\*|do(X^\*))$ is simply the standard observational distribution $P(Y|X^\*)$. This framework establishes the simple semantics—which is agnostic to the LLM's internal mechanisms—as the general and theoretically sound approach for counterfactual reasoning in LLMs.

**Strengths:**

* Theoretical Rigor: The paper comes with rigorous mathematic definitions, theorems, and proofs.
* Potential Practical Improvement: If the claims in the paper hold, the resulting simple semantics would  provide an efficient black-box approach to generate counterfactuals. This is a major advantage over methods that require internal access to the model.

**Weaknesses:**

* **Unconvincing Assumption**:
    * The paper posits that "an LLM can be perfectly described as a causal model," but this stance is  contested by recent work (e.g., the "Causal Parrots: Large Language Models May Talk Causality But Are Not Causal") which argues that current LLMs are not truly causal and are, at best, weak 'causal parrots' that mimic causal language without genuine causal reasoning capability. Could you explain this discrepancy?
* **Dense presentation, leading to an unclear central argument**:
    * The paper is rather dense, riddled with complicated formal statements. Despite spending a decent amount of time, I have a hard time understanding the main argument of the paper. I find the main argument rather range. The key idea of counterfactuals is to use observations to re-shape the distributions of certain random variables, and then predict the outcome of the models based on the updated distributions and the modified input. Under this definition, observing the outcome of an LLM invocation will inevitably affect the internal randomness of the LLM. The paper's main argument is performing counterfactuals can ignore the effect of such observations. I find the argument rather counter-intuitive. Could you explain the intuition behind your main argument informally?

* **Lack of Empirical Validation:** The work is entirely theoretical and lacks any empirical results or case studies. A simple experiment illustrating the practical divergence or convergence of the simple semantics compared to prior methods would significantly strengthen the claims for a machine learning audience.

**Questions:**

Could you please address the two questions raised in "Weaknesses"?

---

> ### Author Response · Authors · 2025-11-15
>
> We thank the reviewer for their  interaction with our paper.
>
> __Unconvincing assumption:__
>
> The reviewer seems to have misunderstood either our paper, or the Causal Parrots paper, because they are both focussing on an entirely different topic. The Causal Parrots paper aims to investigate whether LLMs are good at causal reasoning, and answers this in the negative. Our paper is concerned with representing the LLM itself as a causal model, and that can easily be done: we explain this in our paper, and this is explained in much more detail in Chatzi et al. (2025) and Ravfogel et al. (2025). That such a representation of an LLM can be given, says nothing about the causal reasoning abilities of an LLM, just as a representation of a child’s brain in terms of a neuroscientific model says nothing about the knowledge that a child has of neuroscience.
>
> __Dense presentation:__
>
> That the paper is dense and formally complicated is entirely at odds with the description of reviewer P1G1, who states that our paper is “clearly written and conceptually consistent”, has a “clear and rigorous theory”, is “readable and well organized”, and that our formal results are in fact “straightforward and follow directly from the definitions”. So all we can say is that this is clearly a subjective statement on the part of the reviewer, and that we have done our best to make the paper as clear as possible.
>
> The reviewer also claims that our argument is counter-intuitive, whereas reviewer P1G1 says to the contrary that it “formalizes an intuition already shared by many practitioners”. The fact that both reviewers hold such opposite views, is to us evidence that the truth probably lies in the middle: our approach is intuitive when one just thinks of it informally, in the manner that we explain in the very first paragraph of our paper, but it seems counter-intuitive once one thinks of counterfactuals in the manner that has dominated academic discourse on the topic for the last twenty years, which is to say, from a Pearlian causal modelling perspective.
>
> Unfortunately the only point that both reviewers agree on, is their negative score of our paper...

---

> > ### Comment · Reviewer_x8hu · 2025-11-26
> >
> > Thank you for the response and the clarification on the assumption about LLMs and causality. But my concerns on the representation and empirical validation remain unresolved. Therefore, I am maintaining my score.

---

> > > ### Author Response · Authors · 2025-11-26
> > >
> > > Reviewer P1G1 wants to reject the paper because it makes an entirely obvious point, and the current reviewer wants to reject the paper because they claim that they cannot even understand our point... So in the least, we hope that both reviewers have a discussion on this, because they cannot both be right. The reviewer asks us to explain the main result, but that is exactly what the entire paper does. Could the reviewer please be more specific as to which statements they would like us to clarify?
> > >
> > > Also, we fail to understand the reviewer's concern about empirical validation. Our theoretical contribution results in the view that the counterfactual distribution takes on the same form as the observational distribution. What would it mean to empirically validate that?

---

### Official Review · Reviewer_P1G1 · 2025-10-31

**Soundness:** 4
**Presentation:** 2
**Contribution:** 1
**Rating:** 4
**Confidence:** 3

**Summary:**

This paper models LLMs as nondeterministic causal models and shows that, under their autoregressive structure, counterfactuals collapse to the model’s ordinary predictive distribution. In other words, when we change the prompt from \$x\$ to \$x^\* \$, the counterfactual distribution becomes \$P(Y | X = x^\*)\$; generating a counterfactual simply means prompting the model with \$x^\*\$. The authors contrast this with deterministic causal models, where randomness is shifted into latent variables and counterfactuals become unidentifiable.

The paper is clearly written and conceptually consistent, but entirely theoretical: there are no figures, algorithms, or experiments, and the main contribution formalizes an intuition already shared by many practitioners.

**Strengths:**

Clear and rigorous theory: The nondeterministic causal model formalization and theorems are well presented and logically sound.

Simple but useful baseline: The ''simple semantics'' provides a clean theoretical reference point for future research.

Readable and well organized: The paper is coherent, with arguments that flow clearly and are easy to follow.

**Weaknesses:**

**W1. Incremental contribution:**
    The main claim that simple semantics (i.e., running the LLM again on $x^*$) is a valid way to generate counterfactuals under a nondeterministic causal model is conceptually straightforward and follows almost directly from definitions. The notion of a nondeterministic causal model already exists in the literature, and the relationship between input and output in LLMs naturally satisfies its assumptions. Hence, the main theorem mostly rephrases a known idea rather than offering a substantially new result. In essence, the contribution feels incremental rather than original.

**W2. Philosophical and abstract presentation:**
    The paper reads more like a philosophical essay than an ML paper aimed at a broad audience.  It contains no figures, concrete examples, or empirical illustrations that could make the ideas more accessible to the ML community. Even a minimal diagram of the autoregressive causal structure or a simple experiment would make the work easier to evaluate.

**W3. Limited scope:**
    The result holds only for root-level (prompt) counterfactuals. It is unclear how or whether the same reasoning applies to more general conditioning (e.g., hidden reasoning states or output tokens).

**Questions:**

First, please address the weaknesses mentioned above. I also have two specific questions:

Q1. How should practitioners decide between using simple semantics versus previous methods (e.g., Chatzi et al. [1]) for their applications?

Q2. Is there any way to address the identifiability problems for LLMs in deterministic causal models using Backtracking Counterfactuals [2]?



[1] Ivi Chatzi, Nina Corvelo Benz, Eleni Straitouri, Stratis Tsirtsis, and Manuel Gomez-Rodriguez. 2025. Counterfactual Token Generation in Large Language Models. In Proceedings of the Fourth Conference on Causal Learning and Reasoning.

[2] Von Kügelgen, Julius, Abdirisak Mohamed, and Sander Beckers. "Backtracking counterfactuals." Conference on Causal Learning and Reasoning. PMLR, 2023.

---

> ### Author Response · Authors · 2025-11-15
>
> We thank the reviewer for their interaction with our work.
>
> __W1:__ The reviewer claims that our “main contribution formalizes an intuition already shared by many practitioners”, that it is “conceptually straightforward”, and that the “main theorem mostly rephrases a known idea”. With all due respect, this is demonstrably false. The two approaches we discuss (Chatzi et al. and Ravfogel et al.) are the first and only ones so far to propose a method for generating outputs over counterfactual prompts, and their methods are entirely different from the one we propose. So the approach of our paper does not yet exist anywhere in the literature. Furthermore, if our approach were indeed obvious and shared by many practitioners, then it is hard to explain why these existing papers do not even mention it at all. (See also reviewer x8Hu, who, to the contrary, finds our argument counter-intuitive.)  If the reviewer can refer us to even a single paper that mentions an approach like ours, we will immediately retract our paper.
>
> __W2:__ In their summary, the reviewer first says some positive things, and then finishes with “but the paper is entirely theoretical”. This implies that a paper being theoretical is somehow bad. Similarly, in W2, the reviewer implies that a paper which is partly philosophical, and fails to include figures or empirical results, is somehow bad. We do not understand in the least what justifies such claims. Does the reviewer believe that ICLR is a venue solely for non-theoretical and non-conceptual papers? If so, where is this stated on the ICLR website?
>
> __W3:__ We fully agree that our paper is restricted to discussing only prompt counterfactuals, and we are currently exploring how to expand our semantics to other settings. However, the study of generating counterfactuals for LLMs is still extremely novel, and we here take the very first steps towards offering a theoretical underpinning that can offer a general framework to describe and compare all methods. Without this first step, further steps cannot be taken. For what it’s worth, Chatzi et al. likewise focus exclusively on such counterfactuals.
>
> __Q1:__ That is indeed an important question, and one that should be further explored. The short answer is that it will be highly dependent on what the concrete goal is. For example, as we state in the paper, if one is using counterfactual generation to increase the sample efficiency of comparing the quality of different LLMs, such as in (Benz et al.), then the method of Chatzi et al. is entirely appropriate. On the other hand, if one is interested in using counterfactuals for generating counterfactual explanations, one ought to use either our semantics as is, or construct a novel method that allows for generating counterfactuals that are distant from the actual one. As far as we know, no such method currently exists. We believe that work on such alternative methods was hindered by the lack of a general theoretical framework, and we are thus confident that this will change upon publication of our paper.
>
> __Q2:__
>
>  The connection to backtracking counterfactuals is an interesting suggestion.
>
> First, note that for our simple semantics, and given that we are only considering changing the root prompt (which we agree should be generalized in the future), our semantics can also be seen as a backtracking semantics. This follows from the fact that our semantics agrees with the observational one, and thus the distinction between an interventional and a backtracking semantics disappears. In fact, one can define the simple backtracking semantics as the extreme semantics considered on page 10 of “Backtracking counterfactuals”, where P_B(u* | u)=P(u*). (In our setting, this would be: P_B(x*,y* | x,y) = P(x*,y*).)
>
> Second, the authors of the backtracking paper already make the case that they are useful in the context of generating explanations, for the search for a good counterfactual explanation of some output can be more naturally phrased in abductive terms, i.e., given that we have some actual input and output, what would have been a likely input if the output were different? So instead of having to find or construct a counterfactual input, we reverse the search process, by conditioning on what we want to be the case: that the output would be different. It is not clear whether reversing the search process helps in identifying the ground truth in deterministic LLMs, which is the specific connection the reviewer asks about, but it is definitely useful in the context of self-generated explanations: we can ask the LLM what would be required for it to have answered differently. This offers a different kind of evaluation method for assessing the faithfulness of LLM explanations. In fact, it is this backtracking formulation that is used by Dehghanighobadi et al. (2025).
>
> To conclude, generalizing our framework to include backtracking counterfactuals is another avenue worth pursuing.

---

> > ### Comment · Reviewer_P1G1 · 2025-11-26
> >
> > Thank you for the detailed response. I appreciate the clarifications, but my main concerns remain.
> >
> > I now better understand the intended contribution, and I agree that the perspective is useful. However, once nondeterministic causal models and the specific autoregressive structure of LLMs are assumed, the central result follows fairly directly from these modeling choices. Without additional technical developments, new theoretical insights, or empirical evidence showing practical impact, the overall contribution still feels limited.
> >
> > I also continue to believe the paper would benefit from a concrete illustrative example or a simple diagram to make the ideas more accessible. In addition, the restriction to prompt-level counterfactuals is a meaningful limitation and should be stated more clearly.
> >
> > Overall, while the rebuttal clarified some points, it did not resolve my concerns about novelty, concreteness, and scope. My evaluation remains the same.

---

> > > ### Author Response · Authors · 2025-11-26
> > >
> > > We are unable to understand the reviewer's claim that there is a lack of novelty.
> > >
> > > First of all, nondeterministic causal models and their semantics are themselves very novel, having only been introduced a few months ago, in just two papers. They have not yet been connected to LLMs in _any_ existing work.
> > >
> > > Second of all, the three papers that we engage with are themselves very novel, as they are among the first (probably even the very first) to offer a semantics for counterfactuals of LLMs. This is undoubtedly going to become a very important area of work, given that reasoning about their counterfactual behavior is vital for explanation (as we note in the paper), among other things. These papers all rely _exclusively_ on deterministic causal models, and this has resulted in a very different perspective to counterfactuals than the one we offer. As we stated, if the reviewer is able to find any paper that is even remotely close to suggesting a semantics for counterfactuals of LLMs that is similar to ours, we will retract the paper. (And if not, then what justifies their claim that it "formalizes an intuition already shared by many practitioners"? Where are they?)
> > >
> > > Ironically, the reviewer aims to reject our paper precisely because they believe that the approach is too intuitive and straightforward. If they agree that our approach is on the right track, then surely they should agree with us that it is of the utmost importance that it is published, for it is a matter of fact that all existing approaches conflict with it.
> > >
> > > (As to the point regarding only focussing on prompt counterfactuals: as we stated in our rebuttal, the two papers by Chatzi et al. likewise solely restrict to such counterfactuals.)

---

> > > > ### Comment · Reviewer_P1G1 · 2025-11-26
> > > >
> > > > Thank you for the follow-up comment. Let me clarify my position once more. I am not suggesting that your semantics appears elsewhere or that nondeterministic causal models have previously been applied to LLMs. My concern is about the **magnitude** of the technical contribution. Once nondeterministic causal models and the autoregressive structure of LLMs are assumed, the main result follows quite directly from these assumptions. Because of this, the contribution feels relatively narrow and technically limited, even though the conceptual perspective is interesting and may be useful. In my view, this level of contribution does not fully meet the threshold typically expected for an ICLR main-track paper.
> > > >
> > > > Regarding the “intuition shared by practitioners”, I did not mean to imply that an equivalent **formal** semantics exists. I was referring to the common practical habit of rerunning the model on $x^*$ to see “what the model might have said instead.” Your paper provides a *formal* causal justification for this *informal* practice, which I appreciate, but the derivation is fairly straightforward once the modeling assumptions are fixed. This was the point I intended to make.
> > > >
> > > > I appreciate the authors’ engagement with my comments. I have no further points beyond what is already in my review, and I will leave the final decision to the AC and program committee.

---

### Note · Authors · 2025-12-15

I have read and agree with the venue's withdrawal policy on behalf of myself and my co-authors.